# ELISA with recombinant antigen Lb6H validated for the diagnosis of American tegumentary leishmaniasis

**Ruth Tamara Valencia-Portillo**[1], **José Angelo Lindoso**[1,2,3], **Beatriz Julieta Celeste**[1,4], **Amanda Azevedo Bittencourt**[2,5], **Maria Edileuza Felinto de Brito**[6], **Malcolm Scott Duthie**[7], **Jeffery Guderian**[8], **Jorge Guerra**[9], **Ana Lúcia Lyrio Oliveira**[10], **Steven Reed**[7], **Mussya Cisotto Rocha**[11], **Nicolle Tayná Santos**[9], **Fernando Tobias Silveira**[12,13], **Hiro Goto**[1,4⍟], **Maria Carmen Arroyo Sanchez**[1,4⍟] *

1 Faculdade de Medicina, Instituto de Medicina Tropical de Sao Paulo, Laboratório de Soroepidemiologia, Universidade de Sao Paulo, Sao Paulo, São Paulo, Brazil, 2 Instituto de Infectologia Emílio Ribas, Secretaria de Saúde do Estado de Sao Paulo, Sao Paulo, São Paulo, Brazil, 3 Faculdade de Medicina, Instituto de Medicina Tropical de Sao Paulo, Laboratório de Protozoologia, Universidade de Sao Paulo, Sao Paulo, São Paulo, Brazil, 4 Faculdade de Medicina, Departamento de Medicina Preventiva, Universidade de Sao Paulo, Sao Paulo, São Paulo, Brazil, 5 Global Medical Affairs, MSD in Brazil, Sao Paulo, São Paulo, Brazil, 6 Centro de Pesquisas Aggeu Magalhães, Fundacao Oswaldo Cruz, Recife, Pernambuco, Brazil, 7 Host Directed Therapeutics, Seattle, Washington, United States of America, 8 Access to Advanced Health Institute, Seattle, Washington, United States of America, 9 Fundacao de Medicina Tropical Dr. Heitor Vieira Dourado, Manaus, Amazonas, Brazil, 10 Faculdade de Medicina, Universidade Federal de Mato Grosso do Sul, Campo Grande, Mato Grosso do Sul, Brazil, 11 Laboratório de Investigação Médica (LIM 48), Hospital das Clínicas da Faculdade de Medicina da Universidade de Sao Paulo, Sao Paulo, São Paulo, Brazil, 12 Departamento de Parasitologia, Instituto Evandro Chagas, Pará, Pará, Brazil, 13 Núcleo de Medicina Tropical, Universidade Federal de Para, Pará, Brazil

⍟ These authors contributed equally to this work.
* arroyo@usp.br

## Abstract

American tegumentary leishmaniasis (ATL) diagnosis is an open question, and the search for a solution is urgent. The available tests that detect the etiological agent of the infection are specific for ATL diagnosis. However, they present disadvantages, such as low sensitivity and the need for invasive procedures to obtain the samples. Immunological methods (leishmanin skin test and search for anti-*Leishmania* antibodies) are good alternatives to the etiological diagnosis of ATL. Presently, we face problems with disease confirmation due to the discontinuity in the production of leishmanin skin test antigen, particularly in resource-poor settings. Aiming to diagnose ATL, we validated rLb6H-ELISA for IgG antibodies using 1,091 samples from leishmaniasis patients and healthy controls, divided into four panels, living in 19 Brazilian endemic and non-endemic states. The rLb6H-ELISA showed a sensitivity of 98.6% and a specificity of 100.0%, with the reference panel comprising 70 ATL patient samples and 70 healthy controls. The reproducibility evaluation showed a coefficient of variation of positive samples $\leq$ 8.20% for repeatability, $\leq$ 17,97% for reproducibility, and $\leq$ 8.12% for homogeneity. The plates sensitized with rLb6H were stable at 4˚C and -20˚C for 180 days and 37˚C for seven days, indicating 12 months of validity. In samples of ATL patients from five research and healthcare centers in endemic and non-endemic areas, rLb6H-ELISA showed a sensitivity of 84.0%; no significant statistical difference was observed among the

**Data Availability Statement:** All relevant data are within the manuscript and its Supporting information files.

**Funding:** MCAS received grant #2021/12535-2 São Paulo Research Foundation (FAPESP) (https://fapesp.br/). RTVP received a scholarship from Coordenação de Aperfeiçoamento de Pessoal de Nível Superior – Brasil (CAPES) – Finance code 88882.76665/2019-01 and 88887.689454/2022-00 (https://www.gov.br/capes/pt-br) HG received a research fellowship from Conselho Nacional de Desenvolvimento Científico e Tecnológico – Brasil (CNPq) - n°: 02940/2019-7, (https://www.gov.br/cnpq/pt-br). This study was financed in part by Laboratório de Investigação Médica (LIM 8) Hospital das Clínicas da Faculdade de Medicina da Universidade de Sao Paulo (https://limhc.fm.usp.br/portal/). The funders had no role in study design, data collection, analysis, publication decision, or manuscript preparation.

**Competing interests:** The authors have declared that no competing interests exist.

five centers (chi-square test, p = 0.13). In samples of healthy controls from four areas with different endemicity, a specificity of 92.4% was obtained; lower specificity was obtained in a visceral leishmaniasis high endemicity locality (chi-square test, p<0.001). Cross-reactivity was assessed in 166 other disease samples with a positivity of 13.9%. Based on the good diagnostic performance and the reproducibility and stability of the antigen, we suggest using ELISA-rLb6H to diagnose ATL.

## Introduction

Leishmaniases are neglected diseases caused by protozoa of the genus *Leishmania* prevalent in tropical and subtropical areas and determining the visceral (VL) and tegumentary (TL) forms. The American tegumentary leishmaniasis (ATL) is mainly caused by *Leishmania* (*Viannia*) *braziliensis*, *L.* (*V.*) *guyanensis*, or *L.* (*Leishmania*) *amazonensis*. ATL comprises a broad spectrum of clinical manifestations that depend on the *Leishmania* species, the parasite load, and the host response [1]. Based on the different clinical presentations, it is classified as cutaneous leishmaniasis (CL) in its various forms and mucosal leishmaniasis (ML) [2,3]. World Health Organization estimates that between 600,000 and one million new cases of TL occur annually worldwide, and CL is the most prevalent form [4]. Brazil is endemic for both CL and ML; in 2022, 12,878 cases of ATL were reported [5].

The diagnosis of ATL is based on clinical, epidemiological, and laboratory aspects [3]. Etiological diagnostic methods (based on the detection of parasites or their genetic material) are considered the gold standard; nevertheless, the sensitivity is not high but may increase when the tests are combined [6]. Regarding molecular methods, laboratory standardization is underway [7]. Nevertheless, implementing molecular methods is difficult in settings with limited resources [8].

The immunodiagnosis can be a potential tool to increase access and improve the ATL diagnosis [9–11]. The leishmanin skin test was an alternative used in clinical routines to detect current or past infections [2,3,12]. However, the discontinuity in the production of this antigen has caused problems in diagnosing ATL, mainly in cases of chronic evolution constituting a threat to the patient's health. Without a test for the establishment of the diagnosis, they are often treated based on diagnostic assumptions. Further, the drugs used to treat ATL may cause severe side effects [1,13].

Conversely, serological tests may have advantages for diagnosing ATL, as they use a less invasive sample than skin biopsy and can be automated. They are quantitative and may be easily used at the point of care [14,15]. Assays using recombinant antigens are already integrated into the routine diagnosis of visceral leishmaniasis, and rapid immunochromatographic tests are available for use in patient care settings [16]. However, there is no consensus on the diagnosis of cutaneous leishmaniasis, and serological methods are not incorporated. Since patients with CL produce low levels of anti-Leishmania antibodies [17], there is an interest in developing serological tests that have high sensitivities.

The diagnosis of ATL is an open question in our midst, and the search for a solution is aligned with WHO's goals for leishmaniasis detection, reporting, and treatment [18,19]. In this scenario, a serological test in ELISA format using the recombinant Lb6H to search for anti-IgG antibodies was standardized and validated to diagnose ATL. Furthermore, the option for the rLb6H antigen was based on its performance in a phase 1 study showing 100.0% sensitivity in ATL samples and 98.5% specificity in healthy controls [20]. Therefore, the importance

of this study is evident, as it will fill the lack of a complementary diagnostic test for ATL, and in any case, the more tests available, the faster and more efficient the diagnosis and, consequently, the treatment will be.

## Materials and methods

### Ethics statement

The Ethics Committees that approved this research are 1) Faculdade de Medicina da Universidade de Sao Paulo (FMUSP), Sao Paulo state, approval number 2.530.363. 2) Instituto de Infectologia Emilio Ribas (IIER), Sao Paulo, Sao Paulo state, approval number 3.255.860). The prospective participants signed a written informed consent form. 3) Centro de Pesquisas Aggeu Magalhaes (CPqAM), FIOCRUZ, Recife, Pernambuco state approval number 2.638.758. The participants were obtained retrospectively, and the samples were coded to prevent their identification during or after data collection. 4) Fundacao de Medicina Tropical Dr. Heitor Vieira Dourado (FMT-HVD), Manaus, Amazonas state, approval number 2.584.959. The participants were obtained retrospectively, and the samples were coded to prevent their identification during or after data collection. 5) Hospital das Clinicas da Faculdade de Medicina da Universidade de Sao Paulo (HCFMUSP), approval number 4.717.914. Samples were selected from the SORO-IMT-FM-USP Biorepository. All the samples were coded to prevent their identification during or after data collection.

### Study design

This diagnostic method study employed 1,091 serum samples from ATL patients, healthy asymptomatic individuals, and patients with potentially cross-reactive diseases collected in Brazil's endemic and non-endemic leishmaniasis areas. At Instituto de Infectologia Emilio Ribas (IIER), Sao Paulo, Sao Paulo state, 37 ATL patients were prospectively recruited between 04/21/2019 and 10/28/2021. Centro de Pesquisas Aggeu Magalhaes (CPqAM), FIOCRUZ, Recife, Pernambuco state sent samples from 101 ATL patients retrospectively obtained. The data were accessed on 08/05/2021. Fundacao de Medicina Tropical Dr. Heitor Vieira Dourado (FMT-HVD), Manaus, Amazonas state, sent samples from 67 ATL patients and 32 healthy individuals retrospectively obtained. The data were accessed on 05/06/2021. Eight hundred fifty-four samples were selected from SORO-IMT-FM-USP Biorepository: 258 samples were obtained from ATL patients, 430 from healthy controls, and 166 from patients with other diseases. The data were accessed on 05/20/2021. The experiments were conducted at Instituto de Medicina Tropical da Faculdade de Medicina, Universidade de Sao Paulo, between 2018 and 2021.

### Study subjects

Our cohort consisted of four panels comprising randomized and relabeled samples using study codes to preserve the participants' anonymity and prevent bias. Panel 1 (S1 Fig) consisted of 70 samples of patients with ATL and 70 sera from healthy blood donors selected according to clinical-epidemiological criteria and routine blood bank tests. Panel 2 consisted of 393 samples from patients with ATL, with at least one positive laboratory test at the time of collection, success in the leishmaniasis therapeutic test, or clinical-epidemiological criterion (S2 Fig). ATL samples were collected in five collection centers from three states: Manaus, Amazonas state (N = 63), Recife, Pernambuco state (N = 80), and Sao Paulo, Sao Paulo state (N = 250). The collection centers are reference hospitals that treat patients from different regions. Panel 3 was composed of 392 samples of healthy asymptomatic individuals living in

areas of different endemicities (S3 Fig). The samples of healthy individuals were collected in Manaus, Amazonas state, an endemic area for ATL (N = 32), Tres Lagoas, Mato Grosso do Sul state, an area of VL transmission (N = 90), and two districts in Sao Paulo, Sao Paulo state, one is non-endemic (N = 185), and the other, Engenheiro Marsilac, is a transmission area for CL (N = 85). Panel 3 samples were assayed for Chagas disease and rheumatoid factor, and the 369 samples, negative by both tests, were maintained for the specificity study. For the cross-reactivity study, panel 4 was used (S4 Fig). It was composed of 141 samples from nine different diseases: autoimmune disease (N = 20), Chagas disease (N = 23), histoplasmosis (N = 5), malaria (N = 11), infectious mononucleosis (N = 9), paracoccidioidomycosis (N = 30), syphilis (N = 13), toxoplasmosis (N = 20) and tuberculosis (N = 10), and 25 samples positive to rheumatoid factor.

## Sample size calculation

The calculation of the minimum number of cases and controls to be included in the validation study followed the formula of Banoo et al. [21] and Smith et al. [22], considering sensitivity and specificity of 90%, a lower margin of 95% confidence interval of 85% and a test power of 90%. In this case, a sample size of at least 377 cases and 377 controls was required. In our previous study [20], we obtained a sensitivity of 100.0% (95% CI, 98.3–100.0) in patients with ATL, a specificity of 98.5% (95% CI 92.1–100.0), in healthy individuals, 92.0% (95% CI 87.5–95.3), in patients with other infections, and 93.6% (95% CI 90.1–96.2), in healthy individuals and patients with other infections.

## *Leishmania* recombinant protein

Infectious Disease Research Institute, Seattle, USA (IDRI) gently provided the rLb6H used in this work. This antigen is a heat shock protein identified in 1995 while screening a genomic expression library of *L.* (*V.*) *braziliensis* with serum from a patient infected by *L. braziliensis* with mucosal leishmaniasis and constructed in the ZAP II bacteriophage [23]. In 2017, the protein gene was inserted into the vector pET17b, and for protein expression, the plasmid pET17b-His$_6$-Lb6H was transformed into *Escherichia coli* BL-21 host cells [20]. The rLb6H protein was purified from the insoluble inclusion bodies by affinity chromatography using the nickel-nitrilotriacetic acid (Ni-NTA) protein purification system[20]. The purified protein was evaluated by dodecyl sulfate-polyacrylamide gel electrophoresis (SDS-PAGE) [24], quantified by BCA protein assay [25], and the reactivity against positive and negative sera was analyzed by Immunoblotting [26], at a concentration of 3 μg/mL (18 ng/well).

## Enzyme-linked immunosorbent assay—ELISA

ELISA was performed according to Sato et al. [20] with some modifications. Briefly, 96-well half-area plates (Corning Costar, New York, USA) were sensitized with 50 μL/well rLb6H antigen diluted to 1 μg/mL in 0.06 M carbonate-bicarbonate buffer, pH 9.6. After overnight incubation in a moist chamber at 4˚C, the plates were washed three times with phosphate-buffered saline containing 0.05% Tween 20 (Polyoxyethylene-sorbitan monolaurate, Sigma-Aldrich, St. Louis, EUA) (PBST). Next, the plates were blocked with 125 μL/well of PBST with 5% skimmed milk (PBSTM), incubated in a moist chamber for 2 hours at 37˚C, and washed three times with PBST. After dilution at 1:100 in PBSTM, samples were applied in duplicate (50 μL/ well) and incubated at 37˚C in a moist chamber for 30 minutes, followed by five washes with PBST. Next, anti-human IgG-peroxidase conjugate (Merck 401445, USA), diluted at 1:30,000 in PBSTM (50 μL/well), was added, and the plates were incubated at 37ºC in a moist chamber for 30 minutes, followed by another wash cycle. For reaction development, 50 μL/well of

TMB/$H_2O_2$ (tetramethylbenzidine/$H_2O_2$) chromogen (Novex-Life Technologies, Carlsbad, CA, USA) was added, and the plates were incubated for 10 minutes, protected from light and at 25°C. The reaction was stopped by adding 25 μL/well of 2N $H_2SO_4$. Absorbances were measured in the ELISA reader (Multiskan GO, Thermo Scientific, Finland) at 450 nm. Each plate had positive and negative controls and blank (PBSTL). Sera absorbance values were normalized based on the absorbance of the positive standard serum tested in duplicate on all the rLb6H-ELISA test plates. For each sample, the percentage of the absorbance of the positive standard (ABS%) was calculated:

$$Absorbance\,\%\,of\,the\,positive\,standard = \frac{Sample\,absorbance}{Positive\,control\,absorbance} \times 100$$

## Diagnostic performance of rLb6H-ELISA

The ROC (receiver operating characteristic) curves were constructed using the ABS% values of the samples from panel 1, obtained in rLb6H-ELISA. These ROC curves determined the test's cut-off, sensitivity, specificity, and 95% confidence intervals.

The reactivity index (RI) was calculated by dividing the values of the "ABS%" of each sample by the cut-off obtained in the ROC curve, considering the test reagent when RI $\geq$ 1. The following equation was used:

$$RI = \frac{Absorbance\,\%\,of\,the\,positive\,standard}{cut-off}$$

## Reproducibility, repeatability, homogeneity, and stability of sensitized plates

Five operators performed ten determinations of one positive sample for ATL and one negative on five consecutive days for the reproducibility study. For repeatability, 30 determinations of positive and negative controls were performed on the same plate. The plate's homogeneity was evaluated by applying the positive and negative controls in 15 different plate wells. The coefficient of variation (CV% = standard deviation/ mean of absorbance x 100) was calculated for each parameter.

The stability of sensitized plates was studied by employing 14 plates sensitized with rLb6H (day 0) and storing them at -20ºC (five plates), 4ºC (five plates), and 37ºC (four plates). At -20ºC and 4ºC, the experiments were carried out on days 1, 7, 30, 90, and 180. At 37ºC, the tests were carried out on days 1, 3, 5 and 7. In addition, control plates were sensitized one day before each experiment [27]. Negative and positive controls were assayed in triplicate for the stability test, and a panel of 12 samples from ATL patients with different titers and nine samples from healthy individuals were assayed in duplicate. The stability of the plates at each storage temperature was evaluated by constructing linear regression graphs for each sample tested using the values ABS% obtained at each storage time, both for the test plates and the control plates.

## Validation of rLb6H-ELISA with samples from different regions of Brazil

For validation, panels 2 (ATL patients), 3 (healthy controls), and 4 (other disease patients) were assayed by rLb6H-ELISA.

## Statistical analysis

The results were evaluated using statistical software. GraphPad Prism, version 9.3.1 for Windows (GraphPad Software Inc., San Diego, CA, USA), was used to construct ROC curve and linear regressions; to calculate sensitivity, specificity, positive and negative likelihood ratios with 95% confidence intervals (95% CI); to compare the ABS% obtained with positive and negative samples from Reference Panel using Mann-Whitney Rank Sum Test. R version 4.3.0 for Windows and RStudio version 2023.02.3 [28] for Windows were used to test for the normal distribution of the values (Shapiro-Wilk test) and the homogeneity of variances (Levene test), to construct Venn diagrams, the map, pie charts, and bar graphs. GraphPad/Quickcalcs [29] was used to calculate the confidence interval of proportions, compare two proportions with Fisher's exact test, and calculate the agreement between diagnostic tests and rLb6H-ELISA using the *Kappa* index [30,31]. MedCalc [32] was used to compare the sensitivity of diagnostic tests and rLb6H-ELISA using the McNemar chi-square test and compare three or more proportions with the chi-square test. A significance level of 0.05 was considered ($p < 0.05$).

## Results

### American tegumentary leishmaniasis patient's characterization

Table 1 shows the demographic and clinical data from the ATL patients evaluated in this study.

Of the ATL samples from Panel 1, 64.3% (45/70) were identified by species (Table 2), and 100% had a known clinical presentation. The infecting species was identified through different methods, according to the collection center. At Instituto Evandro Chagas (IEC), Belem, Para state (N = 14), the species was identified by monoclonal antibodies; at Fundacao de Medicina Tropical Dr. Heitor Vieira Dourado (FMT-HVD), Manaus, Amazonas state (N = 4), by PCR–

**Table 1. Demographic and clinical data for American tegumentary leishmaniasis (ATL) patients from endemic areas.**

| Localities | Gender | | | Age (years) | | | Time of disease (months) | | | Clinical presentation | |
|---|---|---|---|---|---|---|---|---|---|---|---|
| | M | F | MI | Median | Min–Max | MI | Median | Min—Max | MI | Cutaneous | Mucosal |
| **Panel 1** | | | | | | | | | | | |
| **Belém, PA–IEC (n = 14)** | 6 | 0 | 8 | 34 | 9–44 | 8 | MI | MI | 14 | 12 | 2 |
| **Manaus, AM—FMT-HVD (n = 4)** | 4 | 0 | 0 | 30.5 | 18–47 | 0 | MI | MI | 4 | 4 | 0 |
| **Recife, PE–CPqAM (n = 21)** | 12 | 9 | 0 | 26 | 17–65 | 0 | 2 | 0.5–3 | 1 | 21 | 0 |
| **Sao Paulo, SP–IIER and HCFMUSP (n = 31)** | 17 | 14 | 0 | 55 | 32–84 | 5 | 12 | 1–636 | 15 | 15 | 16 |
| **Total (n = 70)** | 39 | 23 | 8 | 39.5 | 9–84 | 13 | 3 | 0.5–636 | 34 | 52 | 18 |
| **Panel 2** | | | | | | | | | | | |
| **Manaus, AM—FMT-HVD (n = 63)** | 51 | 12 | 0 | 45 | 20–76 | 0 | 1 | 0.5–120 | 0 | 61 | 2 |
| **Sao Paulo, SP–IIER (n = 136)** | 93 | 43 | 0 | 45 | 7–85 | 3 | 6 | 1–480 | 85 | 77 | 59 |
| **Sao Paulo, SP–HCFMUSP (n = 45)** | 28 | 17 | 0 | 34 | 5–83 | 31 | MI | MI | 45 | 32 | 13 |
| **Sao Paulo, SP–SCMSP (n = 69)** | 45 | 24 | 0 | 56 | 17–84 | 25 | MI | MI | 69 | MI | 11 |
| **Recife, PE–CPqAM (n = 80)** | 61 | 19 | 0 | 28 | 7–66 | 1 | 2 | 0.27–36 | 1 | 79 | 1 |
| **Total (n = 393)** | 277 | 116 | 0 | 40 | 5–85 | 60 | 2 | 0.27–480 | 200 | 249 | 86 |

Panel 1 –reference samples of patients with ATL; Panel 2 –samples of patients with ATL; n–number of samples; MI–missing information; M–male; F–female; Min-Max–minimum-maximum; IEC–Instituto Evandro Chagas, Belem, Para; FMT-HVD–Fundacao de Medicina Tropical Dr. Heitor Vieira Dourado, Manaus, Amazonas; CPqAM–Centro de Pesquisas Aggeu Magalhães, Fundacao Oswaldo Cruz, Recife, Pernambuco; IIER–Instituto de Infectologia Emílio Ribas, Sao Paulo, Sao Paulo; HCFMUSP–Hospital das Clínicas da Faculdade de Medicina da Universidade de Sao Paulo, Sao Paulo, Sao Paulo; SCMSP–Santa Casa de Misericordia de Sao Paulo, Sao Paulo, Sao Paulo.

**Table 2. Samples from patients with American tegumentary leishmaniasis characterized by infecting species (N = 45) used in the rLb6H-ELISA to construct the ROC curve.**

| Infective species | Number of samples | Clinical form | Locality |
|---|---|---|---|
| *L. (V.) guyanensis* | 5 | CL / BDCL | Belém, Manaus |
| *L. (V.) shawi* | 2 | BDCL | Belém |
| *L. (V.) braziliensis* | 32 | CL / ML / DCL | Belém, Recife, Sao Paulo |
| *L (L.) amazonensis* | 6 | CL / ADCL[7] | Belém |

ROC—receiver operating characteristic; CL—Cutaneous Leishmaniasis; BDCL—Borderline disseminated cutaneous leishmaniasis; ML—Mucosal leishmaniasis; DCL—Disseminated cutaneous leishmaniasis, ADCL—Anergic diffuse cutaneous leishmaniasis.

RFLP Hsp70; at Centro de Pesquisas Aggeu Magalhaes (CPqAM), FIOCRUZ, Recife, Pernambuco state (N = 21), by schizodemas; at Instituto de Infectologia Emilio Ribas (IIER), Sao Paulo, Sao Paulo state (N = 6), by restriction enzymes–Hae III (Table 2). At the IIER and the Hospital das Clinicas da Faculdade de Medicina da Universidade de Sao Paulo (HCFMUSP), Sao Paulo, SP, 25 patients were selected by clinical-epidemiological criteria complemented by laboratory diagnosis. ATL patients proceeded from 19 Brazilian states and three neighboring countries (Fig 1).

## Lb6H recombinant antigen reactivity evaluation

The electrophoretic profile in SDS-PAGE is seen in Fig 2A, where the presence of the band corresponding to the rLb6H is seen between the molecular masses of 97 kDa and 66 kDa, with the expected mass of approximately 79 kDa. Fig 2B shows the results obtained in the Immunoblotting; in strip two, a strong band is observed between the high titer leishmaniasis positive control and the band of interest (approximately 79 kDa). In strip three, the reaction with the low titer leishmaniasis positive control was less intense. No band was observed in strip four, which was tested with the negative control, evidencing the absence of reaction.

## rLb6H-ELISA presented high accuracy for ATL diagnosis

The ABS% obtained with samples from the reference panel (Panel 1) showed that rLb6H-ELISA significantly discriminated between healthy controls and ATL patients' samples (p < 0.0001; Mann–Whitney test) (Fig 3A) and demonstrated great accuracy (area under the curve = 1) (Fig 3B). Table 3 shows the diagnostic performance for the rLb6H ELISA, with a sensitivity of 98.6% (95% CI: 92.3–99.9) and a specificity of 100.0% (95% CI: 94.8–100.0) at the cut-off point of 2.8. Considering the ATL patients' clinical form, 74.3% had the cutaneous form, and 25.7% had the mucosal form. One patient with ML caused by *L. braziliensis* was not detected.

## The rLb6H-ELISA meets the requirements for a serological test

The reproducibility, repeatability, and homogeneity were evaluated by calculating the coefficient of variation (CV%) of the absorbances obtained by testing replicates of a positive and a negative serum from panel 1 (S1 Fig), as described above. In the positive samples, the rLb6H-ELISA was reproducible, with a CV% of 18.0% in reproducibility (S5A Fig), 8.2% in repeatability (S5B Fig), and 4.2% in homogeneity (S5C Fig) (S1 Table). These values are within the recommended range (< 25%, < 20%, and < 20%, respectively), confirming the accuracy of the measurements. Regarding the negative samples, CV% values were > 25% in

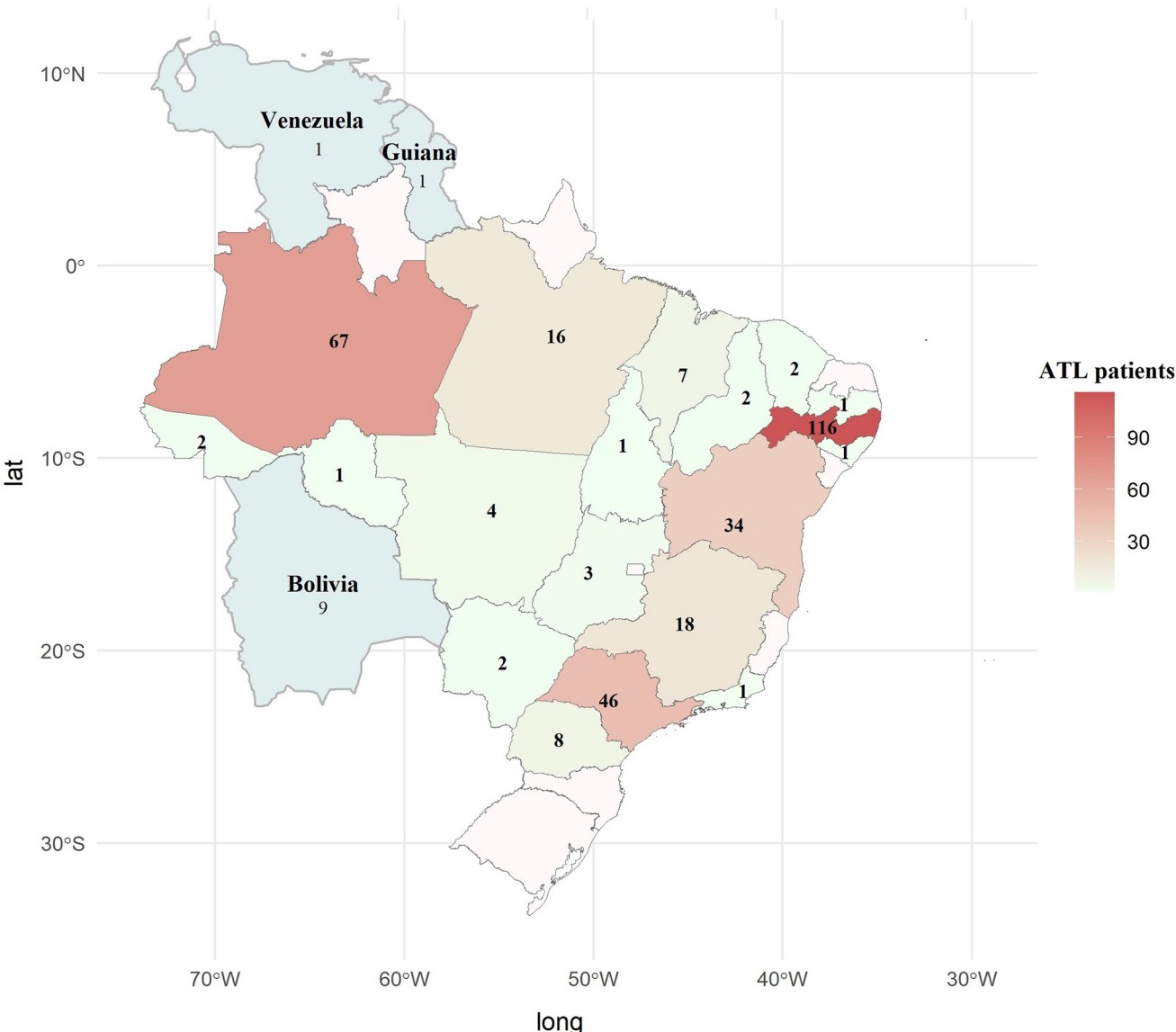

**Fig 1. Geographic distribution of 343 American tegumentary leishmaniasis patients (panel 2) according to the likely site of infection.** ATL–American tegumentary leishmaniasis. The map was generated through RStudio, version 4.3.2, using the geobr package from rstudio.com [33].

reproducibility and > 20% in homogeneity, which may be justified by the increase of imprecision (error) of measurement with the decrease in the concentration of the analyte, leading to a higher coefficient of variation in the repetitions [34]. Despite this, in repeatability, a CV% of 13.9% was obtained.

In assessing the stability of the recombinant antigen on the plate over time, the sensitized plates were stable for 180 days when stored at -20°C and 4°C. Stability at 4°C is advantageous for storing and transporting sensitized plates. In the accelerated stability study, the plates remained stable, without loss of reactivity, for up to seven days of storage at 37°C, indicating the validity for 12 months [27].

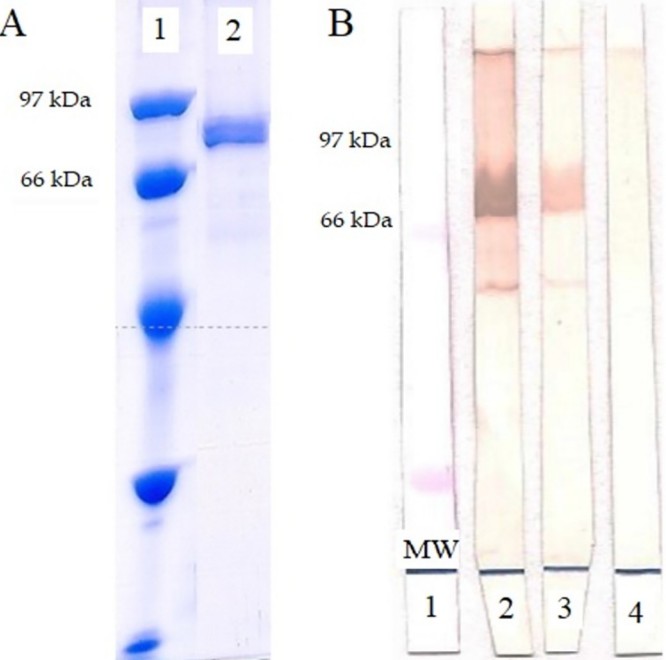

**Fig 2. Evaluation of rLb6H antigen.** A. Electrophoretic profile of rLb6H protein on SDS-PAGE, applying 18 ng of protein per well. B. Immunoblotting reactivity profile of rLb6H. 1- Molecular weight control stained with Ponceau S. 2 —High titer leishmaniasis positive control. 3 –Low titer leishmaniasis positive control. 4—Negative control.

## Validation of the rLb6H-ELISA for the Diagnosis of ATL with samples from different regions of Brazil

After an initial evaluation with the reference panel, the rLb6H-ELISA was validated using panels 2, 3, and 4 (S2–S4 Figs). In the 393 samples from patients with ATL from different regions of Brazil (panel 2), the sensitivity was 84.0% (95% CI: 80,0–87,3). Although the samples came from different endemic areas, similar sensitivity was observed at different collection centers (chi-square test: p = 0.1259) (Table 4).

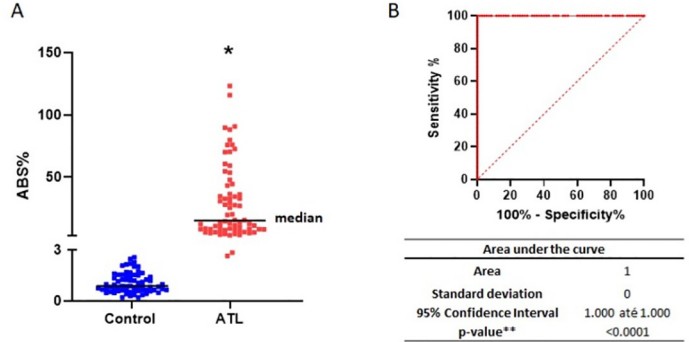

**Fig 3. Performance of rLb6H-ELISA for IgG antibodies in reference panel (Panel 1) (N = 140).** A—Percentage of absorbance of the positive standard (ABS%) obtained in the rLb6H-ELISA; *p < 0.0001 compared to the control, Mann–Whitney test. B—ROC (receiver operating characteristic) curve constructed with ABS% values; ** p-value < 0.0001 means the test significantly discriminates between American tegumentary leishmaniasis patients and controls.

**Table 3. Diagnostic performance of rLb6H-ELISA, using 70 samples from American tegumentary leishmaniasis patients and 70 healthy controls as determined by the ROC curve (Panel 1).**

| Cut-off | N | | | | Diagnostic accuracy (95% CI) | | | | |
|---|---|---|---|---|---|---|---|---|---|
| | TP | FN | TN | FP | Sensitivity % | Specificity % | LR + | LR - | Accuracy % |
| 2.8 | 69 | 1 | 70 | 0 | 98.6 (92.3–99.9) | 100.0 (94.8–100.0) | NC (8.7–∞) | 0.01 (0.008–0.1) | 99.3% (96.1–100.0) |

ROC–receiver operating characteristic. 95% CI– 95% probability confidence interval. TP–true positive. FN–false negative. TN–true negative. FP–false positive. LR+–positive likelihood ratio. LR-–negative likelihood ratio. NC–not calculated.

**Table 4. Sensitivity of rLb6H-ELISA using 393 samples from patients with American tegumentary leishmaniasis (Panel 2).**

| | FMT-HVD (N = 63)[1] | IIER (N = 136)[2] | HCFMUSP (N = 45)[3] | SCMSP (N = 69)[4] | CPqAM (N = 80)[5] |
|---|---|---|---|---|---|
| S % | 93.7 | 83.8 | 75.6 | 84.0 | 81.3 |
| (95% CI) | (84.5–98.2) | (76.5–89.6) | (60.46–87.1) | (73.3–91.8) | (71.0–89.1) |

S -Sensitivity;95% CI– 95% confidence interval;

[1] –Fundacao de Medicina Tropical Dr. Heitor Vieira Dourado, Manaus, AM;

[2] —Instituto de Infectologia Emilio Ribas, Sao Paulo, SP;

[3] –Hospital das Clínicas da Faculdade de Medicina da Universidade de Sao Paulo, Sao Paulo, SP;

[4] –Santa Casa da Misericórdia de Sao Paulo, Sao Paulo, SP;

[5] —Centro de Pesquisas Aggeu Magalhaes, Fundacao Oswaldo Cruz, Recife, PE.

The samples were distributed in four groups, according to the laboratory tests done at the collection centers: immunological diagnosis (IFAT, ELISA, LST) (S6 Fig), etiological diagnosis (direct examination, culture, immunohistochemistry, PCR) (S7 Fig), success in the leishmaniasis therapeutic test and clinical-epidemiological criterion. In summary, 59.0% of the 393 patients had at least one positive etiological test; 68.4% had been positive by at least one immunological test; 2.3% had a positive therapeutic test, and the clinical-epidemiological criterion classified 2.5%.

Regarding the clinical form, 249 (63.4%) patients had cutaneous leishmaniasis, 86 (21.9%) mucosal leishmaniasis, and 58 (14.7%) were classified as tegumentary leishmaniasis. The rLb6H-ELISA showed a significant difference in the percentage of detection of the CL (86.4%; 95% CI: 81.5–90.1) and ML (76.7%; 95% CI: 66.7–84.5) samples, being more sensitive in the detection of antibodies in the cutaneous form (p = 0.0421, Fisher's exact test).

In the analysis of panel 3, the 369 samples that were negative for the presence of rheumatoid factor and anti-*Trypanosoma cruzi* IgG antibodies were considered, and the specificity obtained was 92,4% (95% CI: 89,2–94,7). Across the different collection centers, a significant difference was observed (p < 0.0001, chi-square test), with high specificity in samples from Sao Paulo, Engenheiro Marsilac, Sao Paulo state, and Manaus, Amazonas state (Table 5).

The 166 samples from panel 4 were used in the interference analysis to evaluate the rLb6H-ELISA cross-reactivity, giving 13.9% (95% CI: 9.3–20.0) of positive reactions that exhibited lower-level responses (RI varying from 1.0 to 3.1, and median of 1.7). The diseases that showed the highest proportion of positive reactions were tuberculosis (median positive RI = 2.5), malaria (median positive RI = 1.4), and paracoccidioidomycosis (median positive RI = 1.6) (p = 0.0106, Chi-square test) (Table 6 and Fig 4).

**Table 5. Specificity of rLb6H-ELISA using samples from healthy controls (Panel 3) after the Chagas test[1] and rheumatoid factor test[2].**

|  | Sao Paulo, SP (N = 171) | Tres Lagoas, MS (N = 89) | Engenheiro Marsilac, Sao Paulo, SP (N = 77) | Manaus, AM (N = 32) |
|---|---|---|---|---|
| Sp % (95% CI) | 96.5 (92.5–98.7) | 80.9 (71.2–88.5) | 96.1 (89.0–99.2) | 93.8 (79.2–99.2) |

Sp—specificity; 95% CI– 95% confidence interval;

[1] –ELISA with recombinant antigen (Biolisa, Bioclin);

[2]—Imuno-Látex FR, Wama Diagnostica.

## Evaluation of rLb6H-ELISA as a complementary test for ATL diagnosis

From the 393 ATL patient samples tested by rLb6H-ELISA, 153 were tested by direct exam, 104 by culture, 74 by leishmanin skin test, 51 by immunohistochemistry, 237 by *L. major*-like-ELISA, 282 by immunofluorescent antibody test, and 167 by PCR in the collection centers. Comparing the results of the tests mentioned above with the ones obtained by rLb6H-ELISA (S8 Fig), no significant difference was obtained with *L. major*-like-ELISA (p = 1.000, McNemar test) and PCR (p = 0.1093, McNemar test). The more significant difference was obtained with culture (p = 0.0001, McNemar test), immunohistochemistry (p = 0.0001, McNemar test), immunofluorescent antibody test (p = 0.0010, McNemar test), and leishmanin skin test (p = 0.0371, McNemar test) with a higher positivity by rLb6H-E-LISA (S8 Fig).

Comparing the results, we observed that many positive samples in the rLb6H-ELISA were not detected by the diagnostic tests used in the collection centers. Fig 5 shows the gain in diagnosing the ATL patients with rLb6H-ELISA that varied according to the reference test. Higher gain values were obtained with immunohistochemistry (43%), immunofluorescent antibody test (39%), culture (39%), and direct exam (22%). On the other hand, lower gain values were found with *L. major*-like-ELISA (12%), leishmanin skin test (8%), and PCR (8%). Notably, the high sensitivity of PCR was due to the use of more than one target or different sample types in approximately 50% of patients.

**Table 6. Cross-reactivity of 166 samples from patients with other diseases in the rLb6H-ELISA.**

| Illness | N | Panel 4 | |
|---|---|---|---|
|  |  | N (%) | 95% CI |
| Autoimmune disease | 20 | 0 (0.0) | 0.0–19.0 |
| Toxoplasmosis | 20 | 1 (5.0) | 0.0–25.4 |
| Syphilis | 13 | 0 (0.0) | 0.0–26.6 |
| Paracoccidioidomycosis | 30 | 7 (23.3) | 11.5–41.2 |
| Malaria | 11 | 4 (36.4) | 15.0–64.8 |
| Chagas disease | 23 | 2 (8.7) | 1.2–28.0 |
| Tuberculosis | 10 | 4 (40.0) | 16.7–68.8 |
| Histoplasmosis | 5 | 1 (20.0) | 2.0–64.0 |
| Rheumatoid factor | 25 | 4 (16.0) | 5.8–35.3 |
| Infectious Mononucleosis | 9 | 0 (0.0) | 0.0–34.5 |
| **Total** | **166** | **23 (13.9)** | **9.3–20.0** |

N–number of samples; 95% CI– 95% confidence interval.

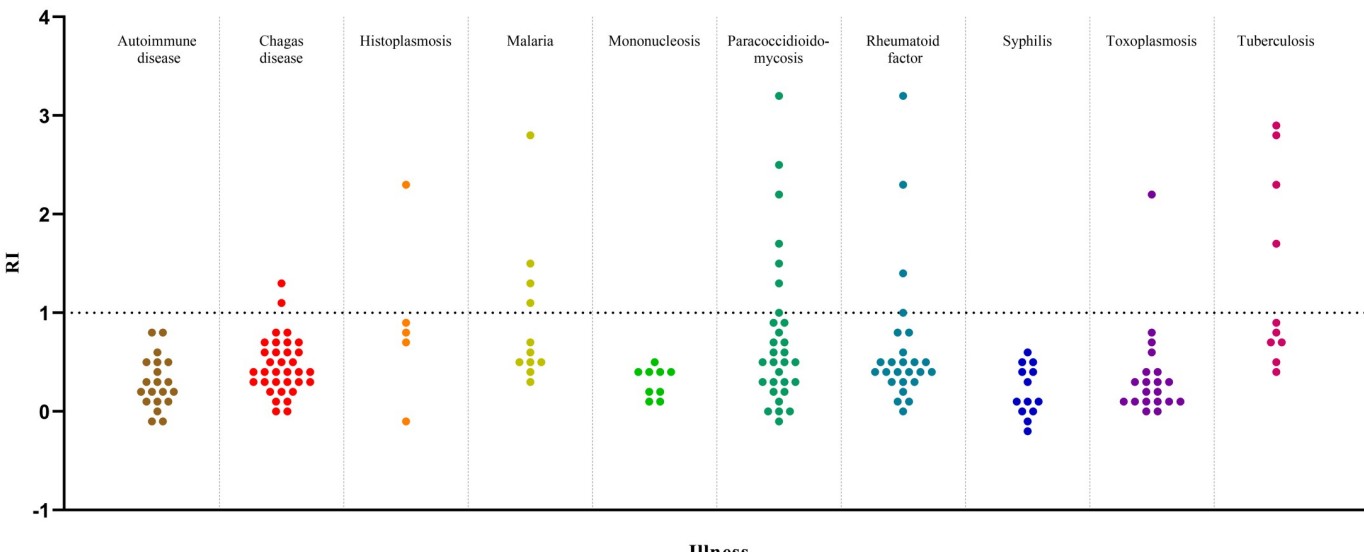

**Fig 4. Reactivity index of samples from other diseases obtained by rLb6H-ELISA RI—Reactivity index.** The black dotted line represents the cut-off.

## Discussion

This study evaluated the performance of the recombinant antigen Lb6H derived from the gene sequence of *L.* (*V.*) *braziliensis* in the ELISA platform for potential application in clinical practice to aid in diagnosing ATL.

The rLb6H-ELISA diagnostic performance obtained in this study was consistent with Sato et al. [20], who used another batch of the same antigen in the phase 1 study. The 95% CI of the positive likelihood ratio (8.7–∞) indicated strong evidence for disease in ATL patients, and the low negative likelihood ratio value (0.01–95% CI: 0.001–0.1) indicated no association with the presence of the disease in healthy individuals [35]. It is worth emphasizing the importance of the likelihood ratio in clinical practice, as it allows the clinician to quantify the probability of disease for any patient. Furthermore, their values do not depend on the prevalence of the disease, as occurs with the positive and negative predictive values, and, therefore, do not vary in different populations [36].

After an initial evaluation with the reference panel, the rLb6H-ELISA was validated using a cohort of samples from cases and controls from 19 Brazilian states and three neighboring countries. The large number of samples from patients with ATL and controls used in the present study represents a significant advantage compared with most studies in Brazil, where a small number of samples are analyzed, mainly from patients with ATL, as described in a recent systematic review [15]. Another important aspect of this study was the detailed characterization of most patients and samples, providing greater robustness to the results. We also selected cases representing the entire spectrum of disease presentation and healthy controls from endemic and non-endemic areas of Brazil to reduce the bias, increasing the sensitivity and specificity of the test [37].

In the diagnosis of cutaneous leishmaniasis, the gold standard tests are specific but lack a high sensitivity, and that is why the collection centers carry out more than one laboratory test to increase the detection of suspected cases. Most ATL-positive samples were collected retrospectively, and it was not feasible to standardize the collection and diagnostic tests performed. Composite reference standards (CRS) increase sensitivity at the expense of specificity unless all

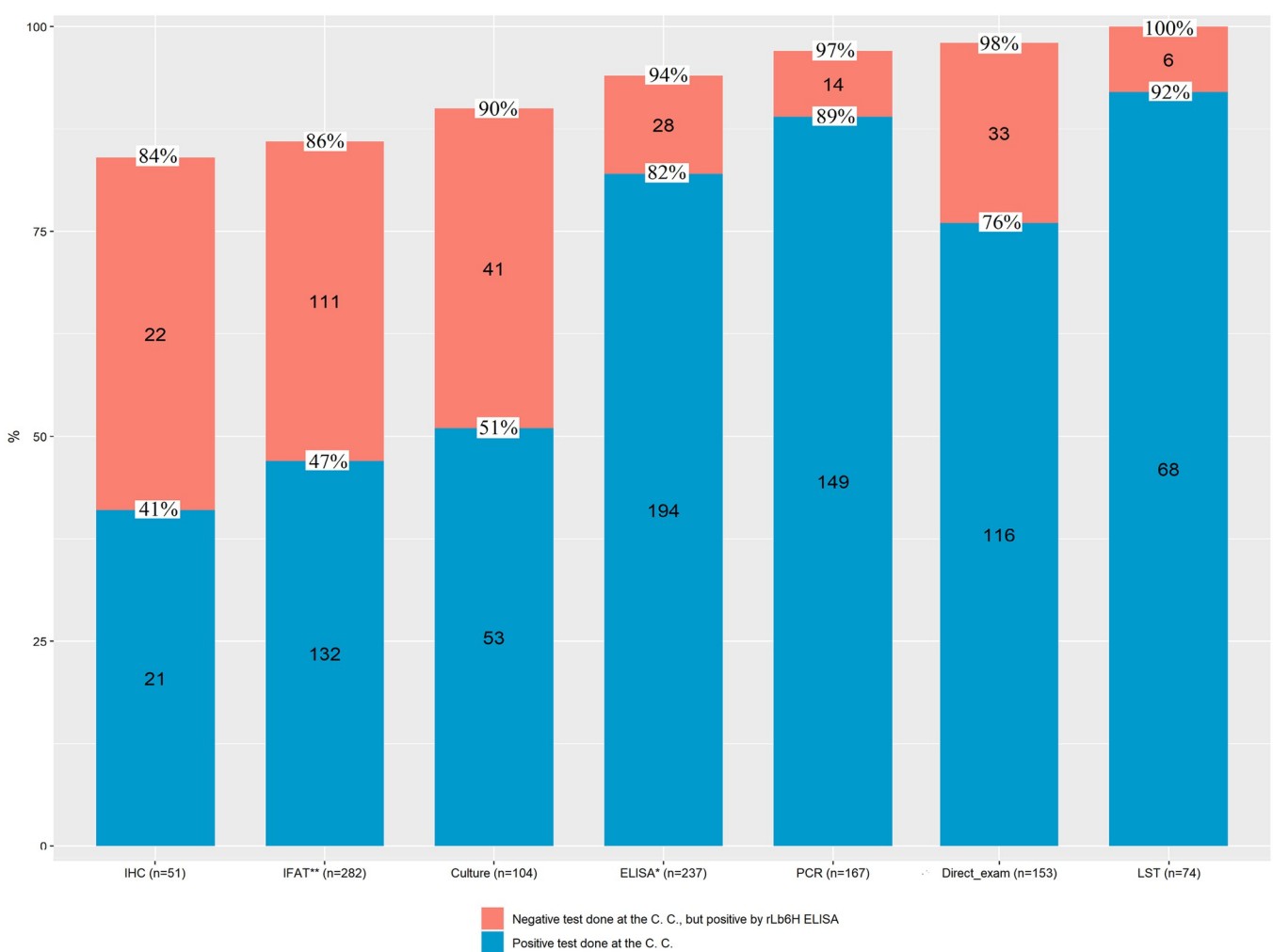

**Fig 5. Diagnostic gain using rLb6H-ELISA as a complementary test.** Blue bars show the number of samples that were positive by the test at the collection centers. Dark pink bars show the number of samples that were negative by the test at the collection centers but positive by rLb6H-ELISA. The percentages of positivity in the collection centers appear between the colored bars. The percentages at the top of the bars refer to the sum of positive samples at the collection centers plus positive samples in the rLb6H-ELISA that were negative by the previous test. IFAT**–immunofluorescent antibody test with *L. major*-like (Sao Paulo) and *L. braziliensis* (Manaus); IHC–immunohistochemistry; ELISA*–enzyme-linked immunosorbent assay with *L. major*-like; PCR–polymerase chain reaction; LST–leishmanin skin test.

component tests have perfect specificity. [38] ATL cases were based on clinical, epidemiological, and laboratory data to reduce bias in using CRS. Although the ATL samples came from different endemic areas, similar test performance was observed regarding sensitivity at different collection centers (p = 0.1259). Regarding clinical forms, rLb6H-ELISA was more sensitive in the detection of antibodies in the cutaneous form (86.4%) compared with ML (76.7%) (p = 0.042). The good performance of the rLb6H-ELISA in patients with CL should be highlighted since these patients produce low levels of anti-*Leishmania* antibodies [17], resulting in variable or low sensitivity of the serological tests [2,39]. Although with a few samples, similar results were obtained in a study with Hsp70 from *L.* (*V.*) *braziliensis* in 30 patients with CL and 20 with ML, with a sensitivity of 83.3% and 85.0%, respectively [40].

In analyzing the 369 samples from healthy controls residing in areas of different endemicities (panel 3), a 92.4% specificity was achieved. Differently from what was obtained with

sensitivity, there was a significant difference in test specificity with samples from different centers (p < 0.0001, chi-square test). Lower levels of specificity were found in Tres Lagoas, MS (80.9%), which is an endemic municipality for VL [41], and it is known that rLb6H may also detect antibodies in the visceral form [20]. Although all participants were asymptomatic, there may be cases of spontaneous cure and subclinical infections, which would be detected as false positives. Although previous studies showed that rLb6H may react with VL samples with a positivity of 89.4% (16), we used another batch of the Lb6H antigen, obtaining 66.7% positivity (Fisher's exact test, p = 0.0025).VL samples (S1 Dataset).

Previous reports have suggested variable sensitivity and specificity using variable recombinant and total antigens for antibody search in ATL. Using *L. braziliensis* Hsp70-ELISA in 50 patients with cutaneous and mucosal forms, a 14% to 84% sensitivity was reported in samples from Cuzco, Peru, and 92% to 100% specificity in 20 sera from residents of Spain and Peru [40]. In Southeastern Brazil, 63% to 94% sensitivity was obtained with *L. braziliensis* Hsp83.1-ELISA in 65 patients with cutaneous and mucosal forms caused by *L. braziliensis* from Belo Horizonte, MG, and 90% to 96% specificity in 50 samples from the non-endemic area of Belo Horizonte [42]. Using total antigen from *L. braziliensis* and *L. amazonensis* promastigotes, ELISA sensitivity ranged from 19% to 81% in 58 samples from Belo Horizonte, MG and Vitória, ES, and specificity, from 58 to 71%, in 49 samples from Uberlandia, MG [43]. With *L. braziliensis* amastigotes, the sensitivity was 98% in samples from endemic areas of Argentina, and the specificity was 98.4% in samples from Japan [44]. The hypothetical protein LiHypA in an ELISA gave 100% sensitivity in 57 ATL leishmaniasis patients and 98.2% specificity in 40 healthy controls and 15 Chagas disease patients [45]. Another hypothetical protein, rLbHyM, was evaluated by ELISA in 45 ATL patients, 50 healthy controls, and 10 Chagas disease patients, with 100% sensitivity and 98% specificity [46]. As stated in the systematic review of Freire et al., among the 38 studies included in the synthesis, most were proof-of-concept and phase I studies [15], with a small number of samples.

The evaluation of potential cross-reactivity of the rLb6H-ELISA showed 13.9% (23/166) of positive reactions with a RI median of 1.7. More positive reactions were obtained in malaria, tuberculosis, and paracoccidioidomycosis patients, corroborating other studies using crude and recombinant *Leishmania* antigens [3,11]. rLb6H-ELISA provided low (8.7%, 2/23) cross-reactivity with Chagas disease, similar to that achieved with rHsp70 (10%), which was the most specific recombinant among those tested by Souza [11]. Conversely, around 100% cross-reactivity with Chagas disease was observed with *L. (V.) braziliensis* amastigotes antigens [44]; 63.0% with antigens of *L. infantum chagasi* promastigotes [11]; 75.8% with *L. major*-like promastigotes [20]. Concerning the cross-reactivity observed with malaria and Chagas disease, the possibility of a latent infection with *Leishmania* cannot be excluded due to the overlapping of endemic areas. Due to the high homology between the sequences of Lb6H and *T. cruzi* heat-shock proteins, one could expect a higher cross-reactivity than obtained in sera from patients with Chagas disease here and by other reports [11,47,48]. However, the possible discontinuity of the similar sequences in the *T. cruzi* protein may give rise to different immunodominant epitopes that elicit antibodies that do not cross-react with rLb6H.

The results of this study, using samples from different regions of Brazil, allow us to state that, in the absence of the LST, the rLb6H-ELISA may be a better alternative for diagnosing ATL since in endemic areas, LST positivity may vary between 20 and 30% in the absence of an active lesion or scar since the test can be positive in cured, asymptomatic individuals, in individuals allergic to the diluent, and with other diseases (Chagas disease, sporotrichosis, multibacillary leprosy, tuberculosis, chromomycosis, among others) [38]. Also, Brazilian studies applying the LST test reported 77,0%-89,1% sensitivity and 60,0%-71,4% specificity [49] similar to or lower than this study's.

One limitation was the diagnostic tests used to characterize ATL patients obtained in different regions of Brazil (panel 2), which were different according to the collection center. Another limitation is the cross-reactivity study (panel 4) due to the lack of samples of cancer patients with skin or oral/nasal mucosa ulcers and the small number of samples of some diseases. The lack of healthy controls from Recife was also a limitation.

The good diagnostic performance, reproducibility characteristics, antigen stability, and validation with samples from endemic areas qualify this rLb6H-ELISA as a good complementary test in diagnosing ATL.

In conclusion, in the scenario where new tests to complement the diagnosis of ATL are needed [6], the rLb6H-ELISA emerges as an option, increasing access availability and establishing a timely and more accurate ATL diagnosis.

## Supporting information

**S1 Checklist. STARD checklist.**
(PDF)

**S1 Fig. Positive and Negative serum samples used to construct the ROC (receiver operating characteristic) curve (Panel 1).** * Diagnosis based on clinical-laboratory criteria. **Diagnosis based on positive parasite culture and *Leishmania* species identification. n = number of samples.
(TIF)

**S2 Fig. Serum samples from American Tegumentary Leishmaniasis patients (Panel 2) obtained in different regions of Brazil.** FMTHVD—Fundacao de Medicina Tropical Dr. Heitor Vieira Dourado, Manaus, AM; CPqAM -Centro de Pesquisas Aggeu Magalhaes, FIOCRUZ, Recife, PE; HCFMUSP—Hospital das Clínicas da Faculdade de Medicina da Universidade de Sao Paulo, Sao Paulo, SP; IIER—Instituto de Infectologia Emilio Ribas, Sao Paulo, SP; SCMSP–Santa Casa de Misericórdia de Sao Paulo, Sao Paulo, SP. n = number of samples.
(TIF)

**S3 Fig. Serum samples from negative control individuals (Panel 3) obtained in different regions of Brazil.** n = number of samples.
(TIF)

**S4 Fig. Serum samples from patients with other diseases (Panel 4).** n = number of samples.
(TIF)

**S5 Fig. Distribution of the coefficients of variations obtained in reproducibility, repeatability, and homogeneity studies.** A–reproducibility, B—repeatability, C–homogeneity. ABS %—Percentage of absorbance of the positive standard.
(TIF)

**S6 Fig. Venn diagrams of samples with a positive immunological diagnosis (N = 269).** IFAT–Indirect Immunofluorescent antibody test; ELISA–enzyme-linked immunosorbent assay; LST–leishmanin skin test.
(TIF)

**S7 Fig. Venn diagram of samples with a positive etiologic diagnosis (N = 223).** IHC–Immunohistochemistry; PCR–polymerase chain reaction.
(TIF)

**S8 Fig. Pie charts show the distribution of positivity obtained by the tests used in each collection center compared with rLb6H-ELISA.** IFAT–immunofluorescent antibody test with *L. major*-like (Sao Paulo) and *L. braziliensis* (Manaus); IHC–immunohistochemistry; ELISA–enzyme-linked immunosorbent assay with *L. major*-like; PCR–polymerase chain reaction; LST–leishmanin skin test.
(TIF)

**S1 Table. Coefficients of variation obtained in reproducibility, repeatability, and homogeneity studies.** 1 –Mean optical densities (OD); 2—SD Standard deviation; 3—CV Coefficient of variation; 4 –P samples from patients with American Tegumentary Leishmaniasis (N = 1); 5 –N samples of healthy individuals (N = 1).
(PDF)

**S1 Dataset. Raw data obtained by rLb6H-ELISA applied to the various groups of samples studied.** ID–identification number; CL–cutaneous leishmaniasis; ML–mucosal leishmaniasis; MCL–mucocutaneous leishmaniasis; ATL–American tegumentary leishmaniasis; Pos–positive; Neg–negative; M–male; F–female; MI–missing information; ND–not done; Age–in years; ELISA binary result: 1– positive; 0 –negative; Evolution time–in months; IFAT–immunofluorescent antibody test; RF–rheumatoid factor; Kalazar Detect and IT-Leish rK39-based RDT: 0 —negative; 1, 2, 3 –positive; DAT–direct agglutination test; RI–reactivity index; VL–visceral leishmaniasis.
(XLSX)

**S1 Raw images.**
(PDF)

## Acknowledgments

The authors thank Taiana Cunha Ribeiro MD, Ícaro Santos Oliveira MD, and Lívia Vieira de Almeida MD for their support in obtaining clinical-epidemiological-laboratory patient data.

## Author Contributions

**Conceptualization:** Ruth Tamara Valencia-Portillo, Hiro Goto, Maria Carmen Arroyo Sanchez.

**Data curation:** Ruth Tamara Valencia-Portillo, Hiro Goto, Maria Carmen Arroyo Sanchez.

**Formal analysis:** Ruth Tamara Valencia-Portillo, Hiro Goto, Maria Carmen Arroyo Sanchez.

**Funding acquisition:** Hiro Goto, Maria Carmen Arroyo Sanchez.

**Investigation:** Ruth Tamara Valencia-Portillo, José Angelo Lindoso, Beatriz Julieta Celeste, Amanda Azevedo Bittencourt, Maria Edileuza Felinto de Brito, Malcolm Scott Duthie, Jeffery Guderian, Jorge Guerra, Ana Lúcia Lyrio Oliveira, Steven Reed, Mussya Cisotto Rocha, Nicolle Tayná Santos, Fernando Tobias Silveira, Hiro Goto, Maria Carmen Arroyo Sanchez.

**Methodology:** Ruth Tamara Valencia-Portillo, Hiro Goto, Maria Carmen Arroyo Sanchez.

**Project administration:** Hiro Goto, Maria Carmen Arroyo Sanchez.

**Resources:** Hiro Goto, Maria Carmen Arroyo Sanchez.

**Software:** Hiro Goto, Maria Carmen Arroyo Sanchez.

**Supervision:** Hiro Goto, Maria Carmen Arroyo Sanchez.

**Validation:** Ruth Tamara Valencia-Portillo, Hiro Goto, Maria Carmen Arroyo Sanchez.

**Visualization:** Ruth Tamara Valencia-Portillo, Hiro Goto, Maria Carmen Arroyo Sanchez.

**Writing – original draft:** Ruth Tamara Valencia-Portillo, Hiro Goto, Maria Carmen Arroyo Sanchez.

**Writing – review & editing:** Ruth Tamara Valencia-Portillo, Hiro Goto, Maria Carmen Arroyo Sanchez.

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
