## [Decision Letter · Decision Letter 0]

6 Feb 2024

PONE-D-23-41150ELISA with recombinant antigen Lb6H validated for the diagnosis of American Tegumentary LeishmaniasisPLOS ONE

Dear Dr. Sanchez,

Thank you for submitting your manuscript to PLOS ONE. After careful consideration, we feel that it has merit but does not fully meet PLOS ONE’s publication criteria as it currently stands. Therefore, we invite you to submit a revised version of the manuscript that addresses the points raised during the review process.

We look forward to receiving your revised manuscript.

Kind regards,

Mariana Lourenço Freire, Ph.D

Academic Editor

PLOS ONE

Journal Requirements:

4. Please amend your authorship list in your manuscript file to include authors Dr. Ruth Tamara Valencia-Portillo, Dr. José Angelo Lauletta Lindoso, and Dr. Beatriz Julieta Celeste. 

5. We note that [Figure 1] in your submission contain [map/satellite] images which may be copyrighted. All PLOS content is published under the Creative Commons Attribution License (CC BY 4.0), which means that the manuscript, images, and Supporting Information files will be freely available online, and any third party is permitted to access, download, copy, distribute, and use these materials in any way, even commercially, with proper attribution. For these reasons, we cannot publish previously copyrighted maps or satellite images created using proprietary data, such as Google software (Google Maps, Street View, and Earth). For more information, see our copyright guidelines: http://journals.plos.org/plosone/s/licenses-and-copyright.

Reviewers' comments:

Reviewer's Responses to Questions

**Comments to the Author**

1. Is the manuscript technically sound, and do the data support the conclusions?

Reviewer #1: Yes

Reviewer #2: Yes

Reviewer #3: Yes

Reviewer #4: Yes

2. Has the statistical analysis been performed appropriately and rigorously? 

Reviewer #1: Yes

Reviewer #2: Yes

Reviewer #3: Yes

Reviewer #4: I Don't Know

3. Have the authors made all data underlying the findings in their manuscript fully available?

Reviewer #1: Yes

Reviewer #2: Yes

Reviewer #3: Yes

Reviewer #4: Yes

4. Is the manuscript presented in an intelligible fashion and written in standard English?

Reviewer #1: Yes

Reviewer #2: Yes

Reviewer #3: Yes

Reviewer #4: Yes

5. Review Comments to the Author

Reviewer #1: The article consists of the validation of an ELISA using the recombinant protein of an hsp of L. braziliensis. The results indicate that the usefulness of this ELISA test is promising. The article is very well written and the results adequately support the conclusions.

However, I have attached some suggestions:

Regarding Lb6H, reviewing the bibliography I must understand that it is the insoluble purification of rLbhsp83a, however, the reference is not clear. It is suggested that this be detailed.

Furthermore, it is suggested that the differences that exist between ATL cases with respect to the sample collection center be discussed.

Reviewer #2: This is an interesting study with use of serum samples from American tegumentary leishmaniasis patients, health individuals and individuals with other diseases from several endemic and non-endemic areas in Brazil, aiming at evaluating the performance of rLb6H-ELISA as a diagnostic tool. Some issues must be addressed by the authors especially regarding the clarity of the titles/legends in the figures and tables.

Figures and tables must be clearly presented and be self-explained, so one is not required to go through the text in order to search for explanations of what is stated. In Table 1, it is not clear what “Panel 1” and “Panel 2” mean. Please briefly define in the title or the legend of Table 1 the meaning of “Panel 1” and “Panel 2”. Similarly, in Table 4 add in the title the total number of ATL patients. In Table 5, explain in the legend the meaning of Sp and 95% CI, and in Table 6 explain in the legend the meaning of 95% CI.

On page 7, lines 137-138: “This study did not include minors.” Table 1 regarding Panel 1 shows age range of 70 patients with ATL from 9 to 84 years, therefore including minors (from Belém-PA and from Recife-PE). Rephrase or withdraw the sentence.

Page 15, line 323, add “titer”: “(…) the high leishmaniasis (…)” = “(…) the high titer leishmaniasis (…). Also in line 324: “(…) the reaction with the low leishmaniasis (…)” = “(…) the reaction with the low titer leishmaniasis (…)”

Page 24, lines 510-513: “The excellent performance of rLb6H-ELISA in patients with CL should be highlighted since the diagnosis is more difficult in this form because there are few parasites in the lesion.” In the context of discussing the best performance of the test in CL (86.4% sensitivity) when compared with ML (76.7% sensitivity), this statement implies that lesions in CL have fewer parasites than those of ML, which frequently is not true. Number of parasites depends on the species of the parasite and clinical and immunological factors from the host, like duration of the disease, presence of disseminated or diffuse leishmaniasis, or comorbidities like HIV coinfection. Consider rephrasing the sentence.

Page 26, line 574: Please change “Wirchowian leprosy” to “multibacillary leprosy”.

Legend of S1 Figure: Please put Leishmania in Italics.

Reviewer #3: The manuscript about ELISA-Lb6H for diagnosing American Tegumentary Leishmaniasis (ATL) is interesting. The results obtained indicate that this method has the potential to be used as a complementary test in diagnosis of ATL. It is also important to highlight the use of samples from different regions of Brazil and three neighboring countries; and cases of ATL caused by different species of Leishmania.

Minor comments

1. Use of "leishmaniases" and "leishmaniasis". Eg, lines 45, 73 and 76. Standardizes the use of "leishmaniasis".

2. S1 dataset: The 21 samples from Recife were identified using Schizodeme. To identify Leishmania species, this method uses parasites, generally isolated by culture. However, for some samples, the authors indicated as "miss information" or negative for cultive and other tests used. How was this identification made?

3. The authors need to improve the discussion on the use of the composite reference standard (CRS) in their study. In line 585 they mentioned this limitation, but it needs to be discussed furhter. Composite reference standards have been advocated in diagnostic accuracy studies in the absence of a perfect reference standard. As the number of component tests increases, sensitivity of this CRS increases at the expense specificity, unless all tests have perfect specificity. Therefore, such a CRS can lead to significantly biased accuracy estimates of the index test. The bias depends on disease prevalence and accuracy of the CRS. Some tests used in the CRS of the present study do not have perfect specificity (eg. IFAT and ELISA). I suggest the following reference on this topic: "Schiller et al., 2015. Bias due to composite referencestandards in diagnostic accuracy studies. Statistic in medicine. DOI: 10.1002/sim.6803".

4. Furthermore, it is necessary to improve the discussion on evaluating the clinical specificity of the ELISA-Lb6H assay. It is important to include in the text the need to evaluate more serum samples from patients with symptoms compatible with CL or ML (eg. skin ulcer or ulcer on the oral and/or nasal mucosa), but with a differential diagnosis that excludes ATL.

Reviewer #4: The manuscript by Velencia-Portillo et al., aimed to carry out a large-scale evaluation of the use of a bacterially expressed, recombinant protein from Leishmania braziliensis for the serological diagnosis of tegumentary leishmaniasis. It builds on previous groups from the same group and others which first assessed the potential use of the same protein for the serological diagnosis of TL. The paper is scientifically sound and is able to evaluate a large number of sera from different sources and also from individuals afflicted by a number of diseases, in order to evaluate cross-reactivity to other diseases. Overall, it strongly supports the use of recombinant proteins for the serological diagnosis of tegumentary leishmaniasis, through ELISA assays, something that has been already clearly stablished for the visceral leishmaniasis, but which has been questioned by some regarding its use for the tegumentary form of the disease. I do find, however, that the manuscript needs a much more detailed description of the antigen it is investigating and why it would be better than others currently assessed for the same purpose. The focus on an antigen that lacks any apparent specificity for tegumentary leishmaniasis should be more clearly explained. Some mention to the serological methods currently used for the visceral leishmaniasis, various commercially available, should also be included.

MAJOR POINTS

The manuscript needs a very thorough revision of the text. The overall organization needs improvement. I find strange, for instance, the large number of paragraphs with single sentences in the Introduction and Discussion. Also, the written English seems in many cases inadequate with many examples where it needs improvement.

I could not find any specific details regarding the recombinant protein used, either in the current manuscript or in the cited references. At least a summary of what it is and what is published regarding it should be mentioned and cited adequately.

I find odd that the authors do not investigate or detail the specificity of the recombinant protein for the different leishmaniasis. There is a mention of it cross-reacting with visceral leishmaniasis, but it is not detailed in any clear form. If this issue has been investigated or is described elsewhere, it has to be detailed. If not, the reasons why also need to be told. If their protein cannot discriminate between different forms of leishmaniasis, this has to be explicitly stated in the title, abstract in the text. For example, the authors can state that their protein can “confirm the diagnosis of leishmaniasis in individuals with suspected ATL”, they cannot confirm that it is indeed ATL, since a person with visceral leishmaniasis would have a positive result as well.

In my view, Tables 1 and 2 and Figure 1 should be included in a first section of the Results, with some of the adjoining text, since a relevant part of the manuscript was devoted to recruiting the various sera groups.

Figure 1 – If the results shown here haven’t been shown before in the previous manuscripts describing the rLb6H recombinant protein, this has to be stated here explaining the need for these experiments at this stage. If similar results have been shown before, there is no need to show them again. Once more, the references to the previous work have to be clearly stated.

Figures 5 and 6 show basically the same result. In my view only one or the other should be shown. The other can be left as supplementary.

MINOR POINTS

Methods – Many paragraphs with single sentences only. In many cases these can be united into a single paragraph.

Lines 83-86 – There is not need for such a detailed description of the distribution of ATL cases in Brazil and the reference in Portuguese is not adequate. Reference “3” would also be inadequate for the same reasons in my view.

Line 88 – Define “Etiological diagnostic methods”

Lines 90-91 – There is no reference to support the following statement “Regarding molecular methods, standardization between laboratories is underway, and their use is still restricted in reference and research centers”.

Line 329 – The total amount of protein loaded on the gel should be stated, not just the concentration used (“ 3μg/mL”).

Line 340 – Clarify what is the unity of the “cut-off point”, whether is the OD value or else.

Lines 362-365 – Figure S5 comes after Figure S1. Confirm if this is OK.

Lines 378-379 – I would expect that the differences in sensitivity observed between FMT-HVD (93.7%) and HCFMUSP (75.6%) would be significant. Please confirm that.

Lines 417-420 – The results showing a greater cross-reactivity with malaria, paracoccidioidomycosis and tuberculosis is noteworthy when the cross-reactivity with Chagas diseases is much lower. The authors should do a better job at discussing why they think this has happened and considering the homology in sequence for the protein between different organisms.

Line 511 – I would not call the performance “excellent”. Please rephrase and tone down.

Lines 479-506 – Please summarize and avoid discussing generalities regarding diagnostic methods and references in Portuguese.

Lines 555-595 – Lots of unnecessary text and information. Please summarize it all in one or two paragraphs. Please tone down the potential use of rLb6H-ELISA as an alternative test as it is, due to the substantial amount of cross-reactivity with other diseases. It is a valid work but as an alternative test some improvement is required.

6. PLOS authors have the option to publish the peer review history of their article (what does this mean?). If published, this will include your full peer review and any attached files.

Reviewer #1: No

Reviewer #2: No

Reviewer #3: No

Reviewer #4: No

---

## [Author Response · Author response to Decision Letter 0]

26 Mar 2024

Dr. Mariana Lourenço Freire

Academic editor

PLoS One

Dear Dr. Freire,

Thank you for carefully analyzing our manuscript, your comments, and suggestions and inviting us to submit a revised version after addressing the suggestions, corrections, and requirements needed for publication at PLoS One.

As asked, we are submitting the files labeled "Response to reviewers", "Revised Manuscript with Track Changes", and "Manuscript". We also resubmit Figure 5, Table 1, Tables 4-6, S8 Figure, S1 Raw Image, S1 Dataset, and STARD-Checklist. We hope that we have addressed all the points raised.

The Funding Information was amended. The financial disclosure is: 

“MCAS received grant #2021/12535-2 São Paulo Research Foundation (FAPESP) (https://fapesp.br/). RTVP received a scholarship from Coordenação de Aperfeiçoamento de Pessoal de Nível Superior – Brasil (CAPES) – Finance code 88882.376665/2019-01 and 88887.689454/2022-00 (https://www.gov.br/capes/pt-br). HG received a research fellowship from Conselho Nacional de Desenvolvimento Científico e Tecnológico – Brasil (CNPq) - n°: 302940/2019-7, (https://www.gov.br/cnpq/pt-br). Laboratório de Investigação Médica - Hospital das Clínicas da Faculdade de Medicina da Universidade de Sao Paulo (LIM 38) (https://limhc.fm.usp.br/portal/). The funders had no role in study design, data collection, analysis, publication decision, or manuscript preparation.”

Kind regards, 

Maria Carmen Arroyo Sanchez 

(on behalf of all authors)

---

## [Decision Letter · Decision Letter 1]

9 May 2024

ELISA with recombinant antigen Lb6H validated for the diagnosis of American Tegumentary Leishmaniasis

PONE-D-23-41150R1

Dear Dr. Sanchez,

We’re pleased to inform you that your manuscript has been judged scientifically suitable for publication and will be formally accepted for publication once it meets all outstanding technical requirements.

Kind regards,

Mariana Lourenço Freire, Ph.D

Academic Editor

PLOS ONE

**Comments to the Author**

1. If the authors have adequately addressed your comments raised in a previous round of review and you feel that this manuscript is now acceptable for publication, you may indicate that here to bypass the “Comments to the Author” section, enter your conflict of interest statement in the “Confidential to Editor” section, and submit your "Accept" recommendation.

Reviewer #2: All comments have been addressed

Reviewer #3: All comments have been addressed

2. Is the manuscript technically sound, and do the data support the conclusions?

Reviewer #2: Yes

Reviewer #3: Yes

3. Has the statistical analysis been performed appropriately and rigorously? 

Reviewer #2: Yes

Reviewer #3: Yes

4. Have the authors made all data underlying the findings in their manuscript fully available?

Reviewer #2: Yes

Reviewer #3: Yes

5. Is the manuscript presented in an intelligible fashion and written in standard English?

Reviewer #2: Yes

Reviewer #3: Yes

6. Review Comments to the Author

Reviewer #2: (No Response)

Reviewer #3: (No Response)

7. PLOS authors have the option to publish the peer review history of their article (what does this mean?). If published, this will include your full peer review and any attached files.

Reviewer #2: No

Reviewer #3: No

---

## [Editor Report · Acceptance letter]

14 May 2024

PONE-D-23-41150R1 

PLOS ONE

Dear Dr. Sanchez, 

I'm pleased to inform you that your manuscript has been deemed suitable for publication in PLOS ONE. Congratulations! Your manuscript is now being handed over to our production team.

Kind regards, 

on behalf of

Dr. Mariana Lourenço Freire 

Academic Editor

PLOS ONE